# Empirical aesthetics of bridges

**Mei Yang** [1]⊙, **Claudia Damiano** [1]⊙*, **Paul Gauvreau** [2], **Dirk B. Walther** [1]

**1** Department of Psychology, University of Toronto, Toronto, Ontario, Canada, **2** Department of Civil Engineering, University of Toronto, Toronto, Ontario, Canada

⊙ These authors contributed equally to this work.
* claudia.damiano@utoronto.ca

## Abstract

Bridges are works of public infrastructure designed to perform a practical function. They are unique among works of engineering in that they also have a significant aesthetic dimension. At their best, they inspire awe and wonder. At their worst, they are eyesores. Little is known about what shapes the aesthetic appeal of bridges. Here we explore how visible features originating primarily from practical considerations relate to aesthetic judgements of bridges. Our dataset comprises of images of 318 bridges from around the world, rated by 254 participants for aesthetic pleasure, interest, complexity, and safety. Civil engineers annotated each bridge's type, depth, visible material, age, and aesthetic premium. Using Factorial Analysis of Mixed Data, we found two significant dimensions. The first dimension, "aesthetics", shows strong correlations among aesthetic, complexity, and interest ratings and is related to bridge type. The second dimension, "perceived safety", relates subjective ratings of safety to bridge age and material. Analyses of visual features, using the Mid-Level Vision Toolbox, shows that contour length and angularity are predictors of the "aesthetics" dimension. For example, cable-stayed bridges are represented by many short and angular contours and are generally rated as more complex, interesting, and aesthetically pleasing. Conversely, slab bridges are often represented by a few long contours and are rated as uninteresting and not aesthetically pleasing. Our study offers the first systematic attempt to collect and analyze subjective ratings of bridge aesthetics, paving the way for empirically supported decisions for the design of bridges and, potentially, other works of public infrastructure.

## Introduction

As elements of urban infrastructure, bridges are not only integral to our transportation networks but are also ubiquitous structures of our everyday landscapes. Bridges are practical objects that exist to perform a useful function. They provide a fixed, above-ground means for vehicles and pedestrians to cross obstacles such as rivers or roads. Bridges also have an undeniable aesthetic dimension. They are

**Data availability statement:** All stimuli and data files are available to researchers via the Open Science Framework (https://osf.io/wf5qg/?view_only=937d90e0271348f6ae313a-1cac2a1062).

**Funding:** This work was supported by a Discovery Grant (RGPIN-2020-04097) from the Natural Sciences and Engineering Research Council of Canada (https://www.nserc-crsng.gc.ca/) and an Insight Grant (435-2023-0015) from the Social Sciences and Humanities Research Council (https://sshrc-crsh.canada.ca/) awarded to DBW. The funders did not play a role in the study design, data collection and analysis, decision to publish, or preparation of the manuscript. There was no additional external funding received for this study.

**Competing interests:** The authors have declared that no competing interests exist.

visible objects and as such, the way they look has the capacity to elicit a subjective response. Like buildings, bridges exist practically everywhere people live and work, and can remain in place for decades. Unlike buildings, bridges invariably stand out from their surroundings as a result of their size, shape, and materials. Some bridges, such as the Golden Gate Bridge in San Francisco or the Ponte Vecchio in Florence, have become iconic tourist landmarks, while others, such as highway overpasses, are overlooked or are even considered unsightly. This variability in both design and appeal underscores the importance of understanding what shapes the aesthetics of bridges. In the current study, we explore the factors that contribute to bridge aesthetics.

Society expects bridges to perform their function in accordance with very high standards of reliability, and for this reason society requires that bridges be designed by engineers. Because most bridges are public works, society likewise expects that their construction will make responsible use of public funds. Engineers respond to these expectations by designing bridges within a strict discipline of economy, which implies no extraneous materials as well as simple and efficient structural systems.

The question of how to design bridges of high aesthetic quality has largely been considered by engineers in terms of how to strike a suitable balance between aesthetic quality and economy. Requirements pertaining to functional reliability, such as those intended to prevent bridges from collapsing, must be complied with without compromise and hence cannot be traded off as a means of optimizing either economy or aesthetics. Regarding the relation between economy and aesthetics, two schools of thought have emerged. The first holds that aesthetic quality cannot be achieved without a commensurate expenditure over and above what would have been required merely to satisfy the practical requirements. This approach is consistent with the truism "you get what you pay for" and has been considered by many throughout the years to be a reasonable basis for design. Given that bridge construction always represents a significant public expenditure, any additional costs incurred to enhance a bridge's aesthetic appeal can place a substantial demand on public funds [1].

The second school of thought holds that economy and aesthetics are not at all incompatible. Scholars of architecture and the fine arts were among the first to point that some bridges that had been designed within a strict discipline of economy, in particular those of the Swiss engineer Robert Maillart, were also works of undeniable aesthetic significance. The writings of Giedion (1952) [2], Mock (1949) [3], and Bill (1955) [4] are notable in this regard. David P. Billington, the first engineer to undertake a systematic study of bridge aesthetics, described in detail the relation between the stresses in a given bridge and its primary visual features [5,6], and in so doing demonstrated convincingly how the visual expression of an efficient flow of forces, conceived within the discipline of economy, can be a rich source of aesthetic expression.

Within both schools of thought, the question of what exactly constitutes a good-looking bridge has been answered exclusively by experts. For example,

some believe that certain types of bridges are inherently more aesthetically pleasing than others [7]. There are also various expert opinions by bridge engineers about what makes a bridge aesthetically appealing, such as certain design choices that support simplicity or minimalism, or using specific materials that promote sustainability and make the bridge more "harmonious with its surroundings" [8,9].

Notably absent is a serious consideration of the opinions of lay people, although they are the ones who will have to live with a given bridge in their visible landscape for years to come and who will pay for any increase in cost resulting from measures added to enhance its visual characteristics. In the rare instances when the opinions of lay people are considered, the methodology often been problematic. One approach [10] involves asking lay people to choose the primary functional features of bridges such as span length. Although span length can certainly affect the appearance of a given bridge, it also has a significant effect on structural behaviour and cost. Lay people do not generally have the specialized training to make an informed decision on these aspects of critical functional features. Another approach involves soliciting the opinion of lay people on a fixed set of design options prepared in advance by experts [11]. Through a suitable selection of the designs within this set, this type of consultation can be implemented to ensure that lay people will end up favouring the design option preferred by the experts. A more general and systematic approach, detached from any specific project, has the potential to yield better insight into how people judge the aesthetic quality of bridges and hence to provide the basis for a better bridge design process.

Nasar [12] recognized that, in the context of urban design, the views of design professionals are often at odds with those of the public. He proposed a general theoretical model of aesthetic response as a basis for resolving this dilemma. According to this model, aesthetic response to a given object is determined by specific visible attributes of the object, together with specific psychological attributes of a given observer. A description of the relation between visible attributes and aesthetic response, which can be obtained empirically, has the potential to provide guidance to design professionals that not only is based on a general understanding of how people judge aesthetic quality, but also is expressed in terms that enable this guidance to be applied directly in design.

Here we apply this approach to bridge aesthetics. Through a collaboration between the fields of visual aesthetics and bridge engineering, our study explores the relationship between a bridge's primary visible features that determine function and cost, hereafter referred to as design features, and its aesthetic appeal. By leveraging our individual expertise, our study presents a pioneering investigation of how specific design features influence aesthetic judgments of bridges.

The design features we explore in the current study span a range of elements. Structurally and visually, bridges can be categorized by their type (e.g., suspension, arch, cable-stayed), visible material (e.g., steel, concrete, wood), apparent age, and other design elements (see Fig 1 for a glossary of design features explored in this study and Methods for more detailed explanations).

In addition, we also relate a more general set of visual properties to the design features and aesthetic judgements of the bridges. This provides a means of examining the results of this study within the context of other important studies of empirical aesthetics. For instance, previous studies have shown that material elements and design details play a role in aesthetic judgements of other architectural structures, such as building exteriors [12,13], indoor spaces [14], urban furniture (e.g., benches) [15], and even automobile interiors [16]. Curvature, in particular, is a feature that is manipulated in art and design and is frequently associated with aesthetic liking. For example, smooth curves are thought to be more pleasing than sharp angles when viewing indoor spaces [17], building exteriors [18], artworks [19], and abstract line drawings [20], among many other stimuli. Importantly, curvature and other contour features such as length and orientation can be computed directly from an image using recently released open-access computational tools. Thus, we also explore whether the design features are associated with specific visual features, and whether these relate to aesthetic judgements similarly to other real-world objects and scenes.

In brief, the current study has two goals. First, to understand the relationship between design features and aesthetic judgements of bridges, and second, to provide the first comprehensive dataset that includes both aesthetic ratings and

**Type:** the classification of a bridge based on its structural design and the method used to support its load

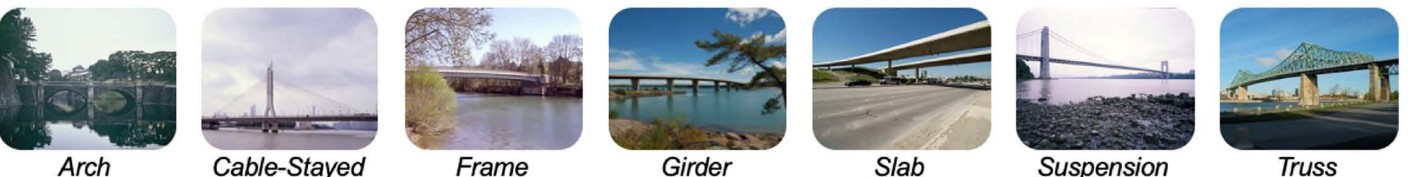

| Arch | Cable-Stayed | Frame | Girder | Slab | Suspension | Truss |

**Depth:** the vertical distance of the primary spanning element of a bridge between two adjacent supports

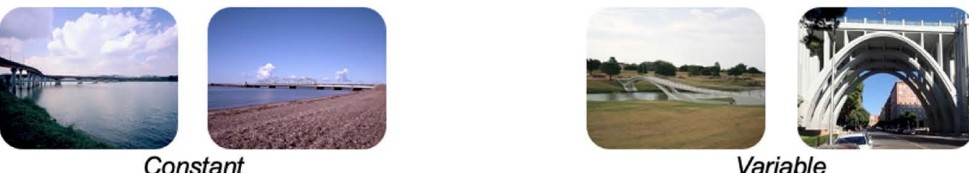

Constant                    Variable

**Visible Material:** the types of construction materials that are externally visible

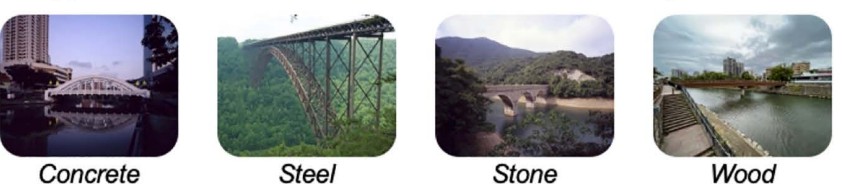

Concrete      Steel      Stone      Wood

**Age:** the perceived age of the bridge based on its visual characteristics

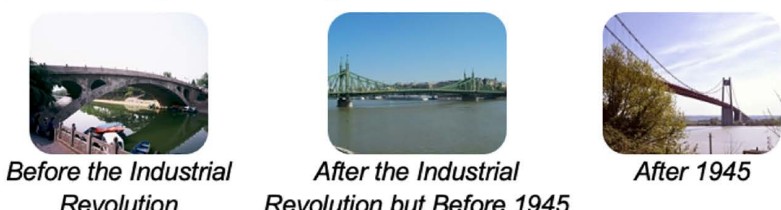

Before the Industrial Revolution | After the Industrial Revolution but Before 1945 | After 1945

**Aesthetic Premium:** the additional design or structural elements beyond basic functional requirements

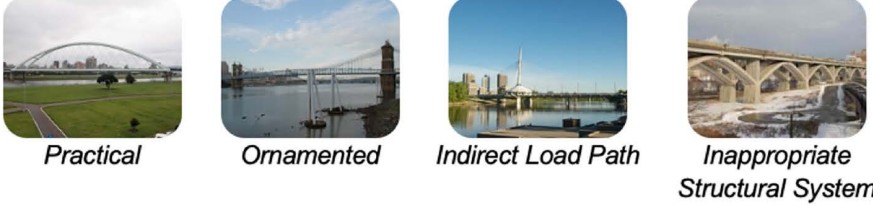

Practical      Ornamented      Indirect Load Path      Inappropriate Structural System

**Fig 1. Glossary of design features along with sample images used in the current study.**

detailed design features of a diverse set of bridges. This foundational study and accompanying dataset aim to serve as valuable resources and inspiration for future research. The results of this work hold the potential to inform the design of future urban infrastructure, ensuring that it meets both practical and aesthetic needs.

## Experiments 1 and 2

### Method

**Stimuli.** The first step towards achieving the objective of this work was to compile two databases of high-quality images of bridges and to annotate each image with a set of design features. Database 1 consists of 491 images of 332 unique bridges. Most of the 332 bridges represented in this database have a single image. The remainder have up to four distinct images. Database 2 consists of four images of 50 unique bridges, for a total of 200 images. The bridges in these databases were selected to provide high-quality images as well as broad diversity in type, scale, age, and location.

The images in both databases consist of colour photographs taken between 1977 and 2023 by one person (author P.G.). All have an aspect ratio of 5 horizontal to 4 vertical. All viewpoints were selected to provide an overall view of the entire bridge, including both superstructure and piers. Views of a given bridge from the perspective of a person travelling across the bridge were therefore excluded. For those bridges represented by more than one image, each image was taken from a significantly different viewpoint. For bridges that incorporate two significantly different structural systems (e.g., a suspension bridge for the main span and a girder system for the approach spans), each system was treated as a separate bridge in the databases.

Each image in the databases was annotated with the name of the bridge and the following features directly related to practical function:

**Type:** arch, cable-stayed, frame, girder, slab, suspension, or truss. This identifies the primary structural system used in a given bridge. The type of bridge determines the primary aspects of its performance under load. Its choice is thus one of the most important decisions made in the design process.

**Depth:** constant or variable. This refers to the thickness of the primary spanning element of a given bridge between two adjacent supports. It is most relevant to bridge types frame, girder, slab, and truss, for which there is a single clearly visible spanning element. Bridge types arch, cable-stayed, and suspension were all annotated as variable depth.

**Visible material:** concrete, steel, stone, or wood. These are the four main materials used in bridge construction from antiquity to the current era. Some bridges incorporate more than one of these materials. For a given bridge, the annotation was made on the basis of which material was most prominent visually.

**Age:** before the Industrial Revolution, after the Industrial Revolution but before 1945, and after 1945. The choice of structural system, depth, and materials has varied over the years, mainly due to advances in technology. The Industrial Revolution and the end of the Second World War in 1945 mark the history of bridges into its three distinct phases. For example, prior to the Industrial Revolution, the only two materials used were stone and wood. Practically all bridges that remain to this day from that era are stone arches. No cable-stayed bridges were built prior to 1945. Annotations were made for the most part on the basis of specific knowledge of a given bridge. Where such knowledge was not available, annotations were made on the basis of knowledge of the types of bridge and materials used in each of the three eras.

Finally, each bridge was also annotated according to the feature **Aesthetic Premium**, which is directly related to economy. Gauvreau [21] proposed a classification of bridges according to whether their design required an expenditure over and above that required merely to perform the required practical function. Bridges for which no such expenditure was required are called "practical" bridges. All other bridges are called "premium bridges", a classification that can be broken down into three categories depending on the specific features that account for the additional expenditure. The first, "ornamented", refers to visual treatments that are added to an otherwise practical structural system. The second, "indirect load path", is a structural system that has been deliberately made inefficient to create a visual impression. Common examples are bridge towers that are tilted rather than vertical. The third, "inappropriate structural system", refers to a structural system that is efficient in its own right but is not the most economical means of performing the practical function. A common example is making spans longer than they need to be to project an impression of grandeur, as would be the case for a bridge that could perform the required function with five spans of 50 metres, but which was designed with a single span

of 250 metres. These annotations were made by an engineering student working under very close supervision from an expert (author P.G.), an experienced bridge engineer, based on extensive knowledge of past and current design practice. The student and expert first annotated several images together, and the expert answered any questions from the student regarding the task. The student then annotated the entire set of images. Following this, author P.G. then reviewed all annotations.

Experiments 1 and 2 both required a dataset with no duplicate bridges. An image set consisting of 118 images corresponding to 118 distinct bridges was therefore selected from the 491 images of Database 1. The selection was made to achieve good diversity of type, location, and age within the constraints of Database 1. All images were resized to 800 by 640 pixels.

Along with the images, we provide a.csv file (e.g., exp1_data.csv) with the ratings of the four scales for Experiment 1 (liking, visual complexity, interest, perceived safety) and five scales for Experiment 2 (liking, visual complexity, interest, perceived safety, familiarity). The rating files also include the design features of the bridges. These design features include the five described above, along with the official name of the bridge, which can also be used to determine the bridge's location. The BDA (Bridge Design and Aesthetics) data set is freely available to researchers via the Open Science Framework (https://osf.io/wf5qg).

## Participants

103 people participated in Experiment 1 (55 men, 40 women, 3 non-binary individuals; $M_{age} = 25.2$ years, $SD_{age} = 3.55$ years, age range = 19–35 years) and 97 people in experiment 2 (43 men, 47 women, 2 non-binary individuals; $M_{age} = 25.2$ years, $SD_{age} = 3.74$ years, age range = 20–35 years). All participants were recruited on Prolific and were paid 1.50 GPB for their participation. Recruitment and testing for Experiment 1 began on July 13th, 2023 and ended on July 21st, 2023. Recruitment and testing for Experiment 2 began and ended on September 20th, 2023. This study was approved by the University of Toronto Research Ethics Board (Protocol number: 30999). We recruited English-speaking Prolific users, from anywhere in the world, that were at least 18 years old, had normal or corrected to normal vision, and had a minimum approval rate of 95%. To ensure that participants were paying sufficient attention to the task, we prompted them to press a certain key twice in the experiment. Participants who failed the attention check trials were excluded from the final sample. Participants who made the same response for more than 40 trials were also excluded. Finally, to ensure that participants finished the experiment in one sitting without rushing through it, we excluded participants whose completion time was three standard deviations away from the mean completion time, which was about 12.16 minutes for experiment 1 and 13.72 minutes for experiment 2 [22]. For Experiment 1, data from 3 subjects were excluded due to failed attention check and 2 subjects were excluded based on completion time. We excluded 5 subjects from Experiment 2 due to failed attention check and 1 subject due to completion time criteria.

## Procedure

Both experiments were programmed using jsPsych [23], a JavaScript framework for creating online psychophysics experiments, and were run on an individually hosted website. To ensure heterogeneous viewing-sequence of images, the image set was pseudorandomly divided into unique halves five times, resulting in 10 subsets of 59 images. 9 to 10 participants viewed each subset in both Experiment 1 and Experiment 2. As a result, we collected 44–54 ratings per image for experiment 1 and 43–48 ratings per image for experiment 2.

At the beginning of the experiment, participants were presented with a digital consent form followed by a brief demographics survey that asked for age and gender. We also collected participants' background via the following items: (a) Do you have a background in architecture? (b) Do you have a formal education in architecture? (c) Do you have a background in engineering? (d) Do you have a formal education in engineering? Participants were given the option to choose between 0 = yes and 1 = no.

Participants were then given instructions that they would be viewing images of bridges and would be asked to provide ratings for them. Each trial started with an image in the center of the screen on a light gray background, and the rating scale was presented beneath the image (see Fig 2). Participants were then asked to respond on a 5-point Likert scale via mouse click to the following statement: "I find the design of the bridge to be aesthetically appealing." The response options were 1 = strongly disagree, 2 = disagree, 3 = neutral, 4 = agree, 5 = strongly agree. After participant made a response, the image would remain the same, but the scale will be replaced by the next rating item. The subsequent rating items are complexity ("I find the design of the bridge to be complex"), interest ("I find the design of the bridge to be interesting."), and perceived safety ("I would feel safe crossing this bridge."). The response options to all these items are identical, with 1 = strongly disagree and 5 = strongly agree. For Experiment 2, participants additionally had to indicate their familiarity with the bridge ("Do you recognize this bridge?"). The response options were 0 = yes and 1 = no. Although we did not limit participants' viewing times, they were instructed to try and respond within 3000 ms for each rating item. After participants completed all rating scales for one image, the process was repeated until all 59 images had been rated.

Participants were given a break every 76–95 trials and were instructed to press any key to continue the experiment. After the main experiment, participants were asked to indicate if they had encountered any technical difficulties, such as failed image loading and low image quality. No such technical difficulties were reported.

## Data analysis

We computed the average ratings of aesthetic liking, complexity, interest, and safety based on participants' responses in the task. The ratings were firstly z-normalized for each participant and each task. Then, for each image, we averaged

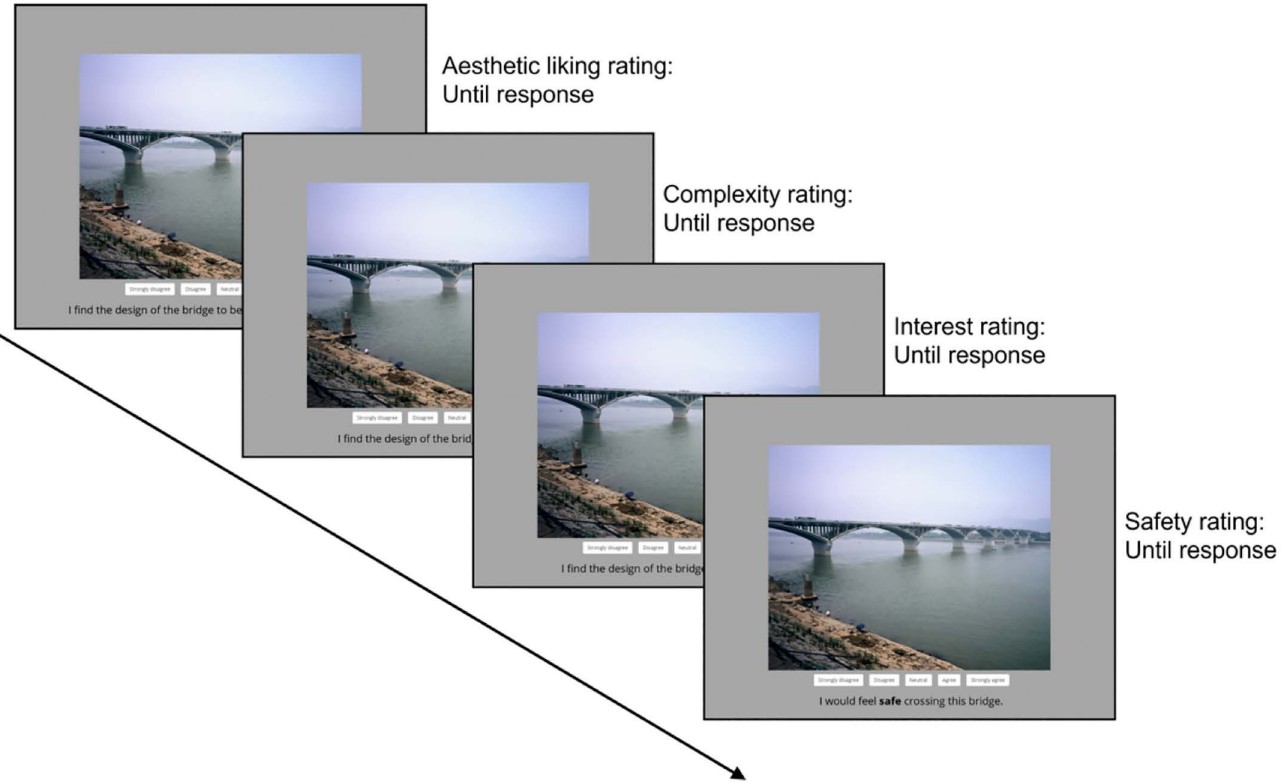

**Fig 2. Example trial sequence for Experiments 1 and 2.** From top to bottom are rating scales of aesthetic liking, complexity, interest, and perceived safety.

across all participants who viewed that image, resulting in one rating per image for each of the subjective measures. For each rating item, we excluded outliers by removing trials where the aggregated score was more than three standard deviations away from the mean. As a result of this criterion, we removed trials containing one safety outlier from the rating data. For Experiment 2, we additionally removed trials containing recognized bridge images by each participant. We then merged the rating data with the design features to be used for main analysis.

The presence of multiple continuous and categorical variables in our data necessitates the use of a multivariate statistical analysis. To that end, we employed Factorial Analysis of Mixed Data (FAMD) to understand the relationship between the subjective ratings of bridges and their design features [24,25]. FAMD is a type of dimensionality reduction technique suitable for datasets containing both continuous and categorical variables. It can be considered as a combination of principal component analysis (PCA) and multiple correspondence analysis (MCA). Specifically, FAMD starts by converting categorical variables into indicator matrices and combines them with the standardized continuous variables. Then, FAMD performs PCA on the transformed data to determine principal components. PCA identifies the directions (principal components) along which the variance in the data is maximized, thereby reducing redundancy and highlighting the most important features of the data. This is particularly important when quantitative variables are highly correlated (see S3 and S4 Figs). In addition to the usual benefits of dimensionality reduction, FAMD is particularly useful in our analysis for several reasons. First, it enables us to visualize and compare the contribution of each variable to the overall variance within the data. Second, FAMD facilitates meaningful comparisons both among and within categorical and continuous variable sets while preserving their original data types. Third, by representing each observation on the factor map, FAMD allows for comparisons among individual bridges. All data analysis was conducted using the R statistical software (R version 4.1.3). FAMD was performed using the FAMD function from the FactoMineR package [26].

## Image feature analysis

In order to extract the contour features of the bridges only, we first had to isolate them from their backgrounds. To do so, we used Meta's Segment Anything algorithm [27]. For each image, we uploaded it to Segment Anything, clicked on the bridge portion of the image to extract the relevant segment, and saved only that segment as a new image. If a bridge could not be easily isolated with this algorithm, because of background clutter or unclear contour boundaries, we manually extracted the bridge using DataLoop (dataloop.ai) by clicking along the boundaries of the bridge, creating a mask, and using that mask to extract the bridge from the original image to save it as a new image. Only 11 of the 118 bridges had to be manually extracted in this way. In the end, each bridge was saved as a PNG image with transparent background. We then ran the isolated bridge images through the MLV toolbox [28] to extract line drawing versions of each bridge (see Fig 3).

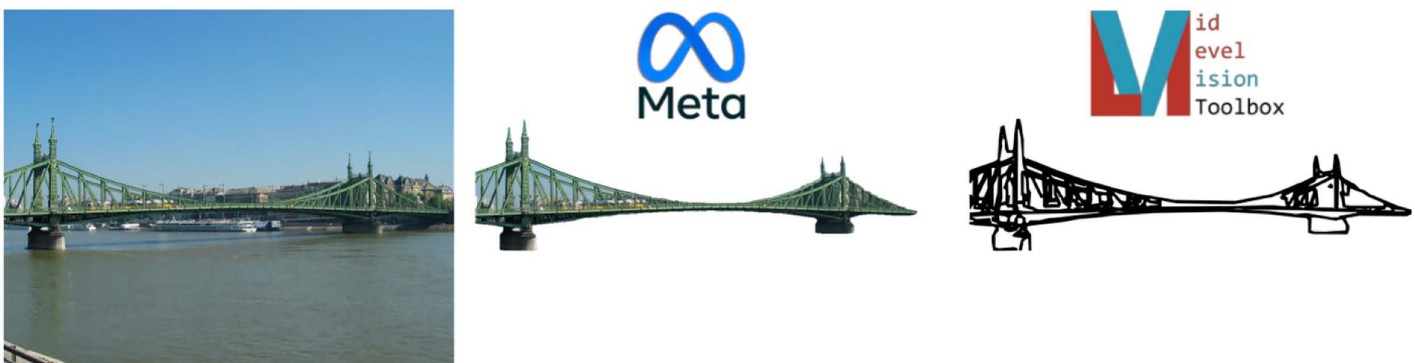

**Fig 3. Bridge extraction pipeline for image feature analysis.**

From the line drawings, we can calculate the length, orientation, curvature, and junction features. These features have previously been found to be important for scene perception [29], as well as for emotional judgements of scenes [20]. We additionally extracted global features from the background of each image to check whether any features that were not related to the bridge itself contributed to our findings. To do so, we first masked the portion of the image that contained the bridge, then computed the following global features: proportion of green pixels, proportion of blue pixels, and Shannon entropy, all of which have been previously associated with aesthetic pleasure [30–32]. The final feature we calculated was the proportion of pixels taken up by the bridge within each image, to determine whether the depicted size (not real-world size) of the bridge influenced aesthetic ratings.

Finally, to determine whether certain image features were associated with subjective ratings of pleasure, interest, complexity, and perceived safety, as well as with the annotated design features, we extracted each image's Dimension 1 and 2 weights from the FAMD analysis and related them to the visual features. Specifically, we ran two separate multiple linear regressions using each image feature (described above) as predictors for the dimension weights.

## Results

### Experiment 1

We conducted an FAMD on the data from Experiment 1 that contained five design features (type, depth, aesthetic premium, visible material, age) and four subjective ratings (aesthetic liking, complexity, interest, safety). The FAMD found five principal dimensions that cumulatively explain 55.03% of the variance within the original data (see Table 1). We retained the first two dimensions, as they cumulatively explain 31.31% of the variance and that there is a clear elbow point on the scree plot (see S1 Fig) after the second dimension. The factor map (Fig 4) reveals that aesthetic liking, complexity, interest ratings, and bridge type are major contributors to the first dimension, whereas material, apparent age, aesthetic premium category, and perceived safety ratings contribute primarily to the second dimension. This divergent pattern of clustering led us to name the first dimension as the *Aesthetic* dimension and the second dimension as the *Perceived Safety* dimension. Fig 5 highlights the most contributory variables to the Aesthetic and Perceived Safety dimensions, and the full table of contribution can be found in the supplementary (S1 Table). Bridge type significantly contributes to both the *Aesthetic* (17.4%) and *Perceived Safety* (17.2%) dimensions, whereas the aesthetic premium category minimally contributes to the *Aesthetic* dimension (4.89%) and moderately contributes to the *Perceived Safety* dimension (12.27%). A further breakdown of the design features (Fig 6) indicates that certain bridge types such as cable-stayed, suspension, and truss bridges positively contribute to the Aesthetic dimension, and slab, girder, and frame bridges negatively contribute to the aesthetic dimension. For the Perceived Safety dimension, stone bridges and bridges made before the industrial revolution were found to positively contribute to the Perceived Safety dimension, whereas bridges for which the aesthetic premium category was Indirect load path or Inappropriate structural system were found to negatively correlate with the Perceived Safety dimension.

### Experiment 2

Familiarity has been found to contribute to aesthetic preferences potentially via processing fluency [33]. To control the effect of familiarity on participants' aesthetic responses to bridges, we additionally collected participants' familiarity ratings

**Table 1. Eigenvalue and % total variance for principal components of Experiment 1.**

| Dimension | Eigenvalue | % of Variance | Cumulative % |
|---|---|---|---|
| 1 | 4.02 | 21.13% | 21.13% |
| 2 | 1.93 | 10.18% | 31.31% |
| 3 | 1.67 | 8.81% | 40.12% |
| 4 | 1.55 | 8.15% | 48.27% |
| 5 | 1.28 | 6.76% | 55.03% |

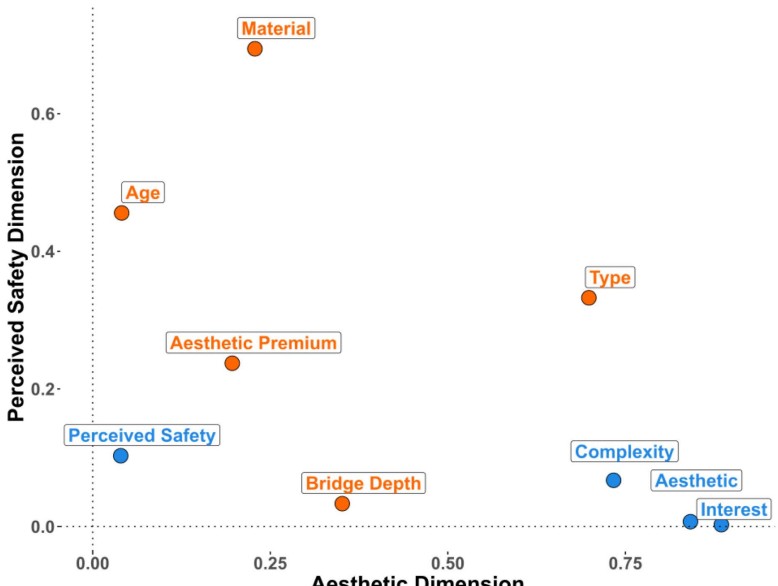

**Fig 4. Variable factor map resulted from factorial analysis of mixed data (FAMD) for Experiment 1.** FAMD combines techniques of Principal Component Analysis (PCA) for continuous variables and Multiple Correspondence Analysis (MCA) for categorical variables. Aesthetic dimension (Dimension 1) predominantly features contributions of interest, aesthetic, and complexity rating. Perceived Safety dimension (Dimension 2) mainly features contributions of material, age, bridge type, and aesthetic premium category.

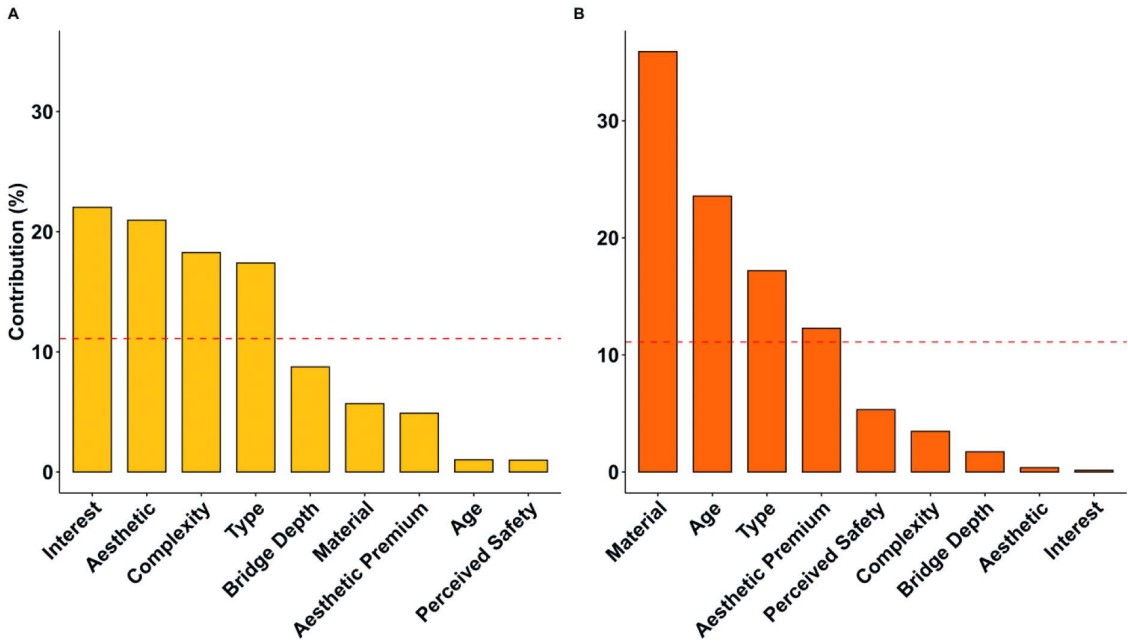

**Fig 5. Bar chart of the most contributary variables to the aesthetic and perceived safety dimension in Experiment 1.** The reference dashed line corresponds to the expected value if the contributions were uniform. (A) shows that interest, aesthetic liking, complexity rating and bridge type are highly contributary to the Aesthetic dimension. (B) shows that material, age, bridge type, and aesthetic premium are highly contributary to the Perceived Safety dimension.

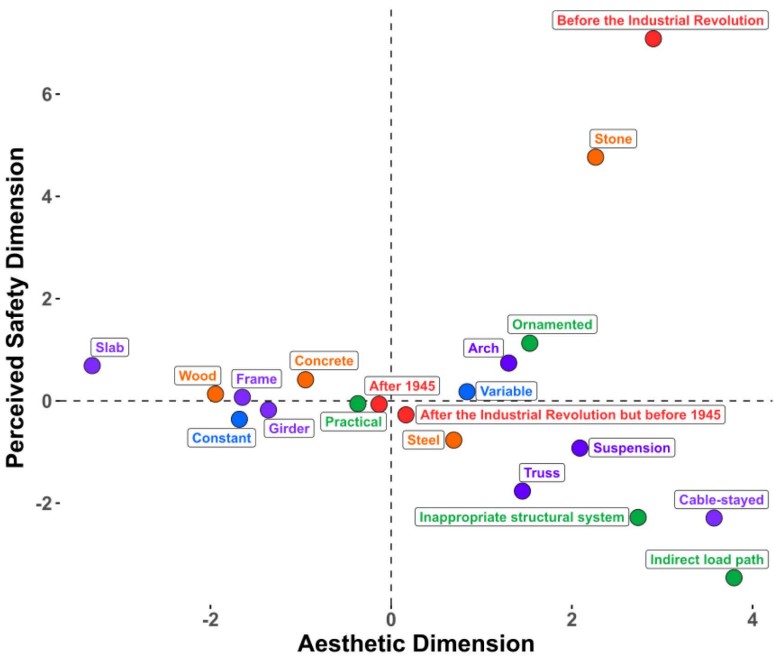

**Fig 6. Contribution of design features to the aesthetic and perceived safety dimensions of Experiment 1.** Variables are colored based on their parent group, with each point representing a specific design feature. Bridge type is represented in purple, depth in blue, visible material in orange, aesthetic premium in green, and age in pink.

of bridges. Although most bridge images were unrecognized by participants (see S5 Fig), we still excluded individual trials that were rated as familiar for each participant from the final analysis. Then, we conducted FAMD and multiple linear regression as a replication of Experiment 1. FAMD found a total of five principal components capturing a cumulative explained variance of 54.67% (Table 2). We chose the first two dimensions, which cumulatively explain 31.23% of the variance within the data (see S2 Fig for the scree plot). Fig 7 presents the factor map, where we replicated the clustering of "aesthetic" variables and "perceived safety" variables. The contributions of variables to each dimension can be more clearly inspected in Fig 8, where we find that interest, complexity, aesthetic pleasure, and bridge type are major contributors to the *Aesthetic* dimension, just as we found in Experiment 1. Material, age, bridge type, and aesthetic premium contribute to the *Perceived Safety* dimension. Contribution of specific design features to each dimension also successfully replicates Experiment 1 (Fig 9 and S2 Table). Cable-stayed, suspension, and truss types are found to positively correlate with the *Aesthetic* dimension, whereas slab, frame, and girder bridge types are found to negatively correlate with the *Aesthetic* dimension. Stone bridges and bridges built before the Industrial Revolution positively correlate with the *Perceived Safety* dimension, whereas bridges with the aesthetic premium aesthetic category of inappropriate structural system and

**Table 2. Eigenvalue and % total variance for principal components of Experiment 2.**

| Dimension | Eigenvalue | % of Variance | Cumulative % |
|---|---|---|---|
| 1.00 | 4.00 | 21.03% | 21.03% |
| 2.00 | 1.94 | 10.21% | 31.23% |
| 3.00 | 1.68 | 8.84% | 40.07% |
| 4.00 | 1.53 | 8.07% | 48.15% |
| 5.00 | 1.24 | 6.53% | 54.67% |

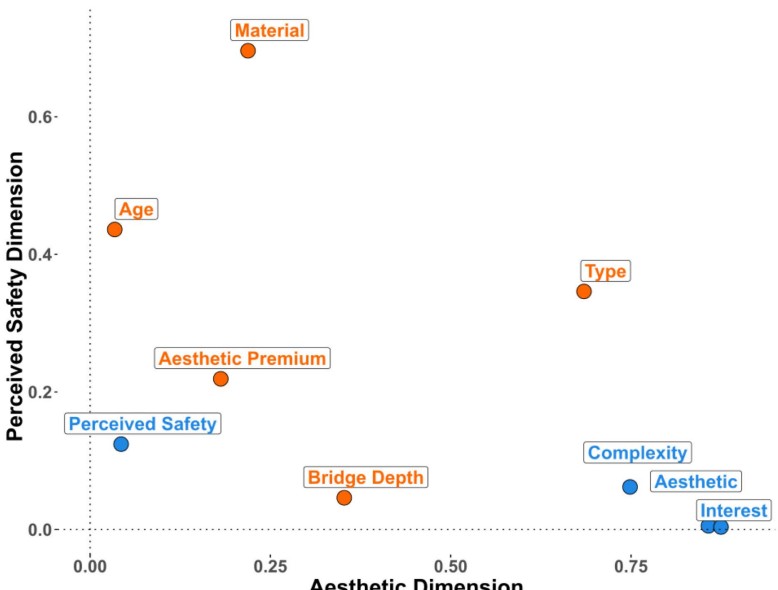

**Fig 7. Variable factor map resulted from FAMD for Experiment 2.** Aesthetic dimension (Dimension 1) predominantly features contributions of interest, aesthetic, and complexity rating. Perceived Safety dimension (Dimension 2) mainly features contributions of material, age, type, and aesthetic premium.

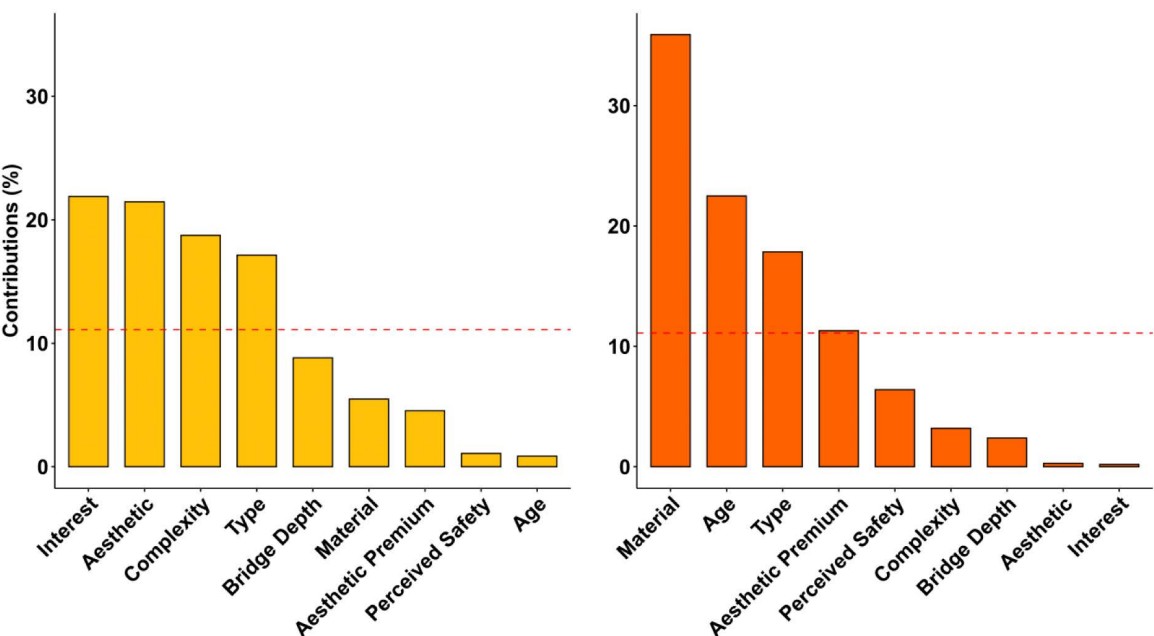

**Fig 8. Bar chart of the most contributary variables to the aesthetic and perceived safety dimension in experiment 2 replicates patterns in Experiment 1.** (A) shows that interest, aesthetic liking, complexity rating and bridge type are highly contributary to the Aesthetic dimension. (B) shows that material, age, type, and aesthetic premium are highly contributary to the Perceived Safety dimension.

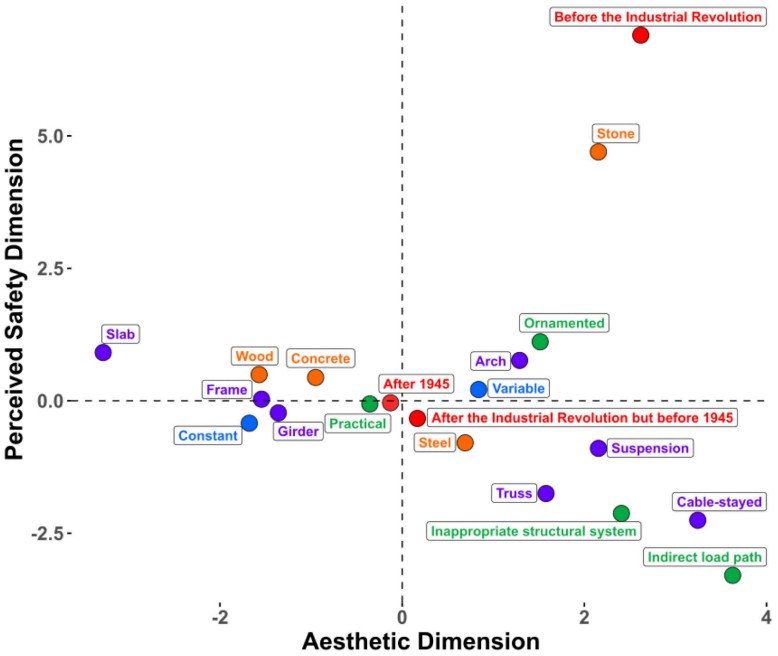

**Fig 9. Design features' contribution to the aesthetic and perceived safety dimension of Experiment 2.** Variables are colored based on their parent group, with each point representing a specific design feature. Bridge type is represented in purple, depth in blue, visible material in orange, aesthetic premium category in green, and age in pink.

indirect load path are found to negatively correlate with the *Perceived Safety* dimension. Overall, the results of Experiment 2 replicate Experiment 1, suggesting that people can consistently and reliably evaluate various aspects of bridge design.

## Image feature analysis

Dimension weights were extracted from FAMD analysis performed on combined data from Experiments 1 and 2. A multiple linear regression, predicting the weights of each image on Dimension 1 (the *Aesthetic* dimension) from the image features, found a significant relationship ($F(11,105) = 2.87$, $p = 0.002$, adjusted $R^2 = 0.15$). Of all image features included, only length ($\beta = -0.02$, $p = 0.009$) and angularity ($\beta = 0.73$, $p = 0.02$) contributed significantly to the model. Results revealed that the Aesthetic dimension was positively related to angularity and negatively related to length (see Fig 10), meaning that bridges with shorter and more angular contours on average were rated as more pleasing, interesting, and complex, and were associated with specific bridge types such as cable-stayed and suspension bridges, while bridges with longer flatter contours, such as slab, girder, and frame bridges, were found to be less pleasing, interesting, and complex. No other features were significantly predictive of this dimension (all $ps > 0.05$).

Dimension 2 (the *Perceived Safety* dimension) was not predictable from any of the extracted visual features ($F(11,105) = 1.78$, $p = 0.07$, adjusted $R^2 = 0.07$), suggesting that subjective safety, as well as visible material and age, are not related to contour features.

## Experiment 3

### Methods

**Stimuli.** Given that only contour features of the bridge, and not other features of the background, were predictive of the Aesthetics dimension, we assume that this meant participants followed instructions and made their judgements on the

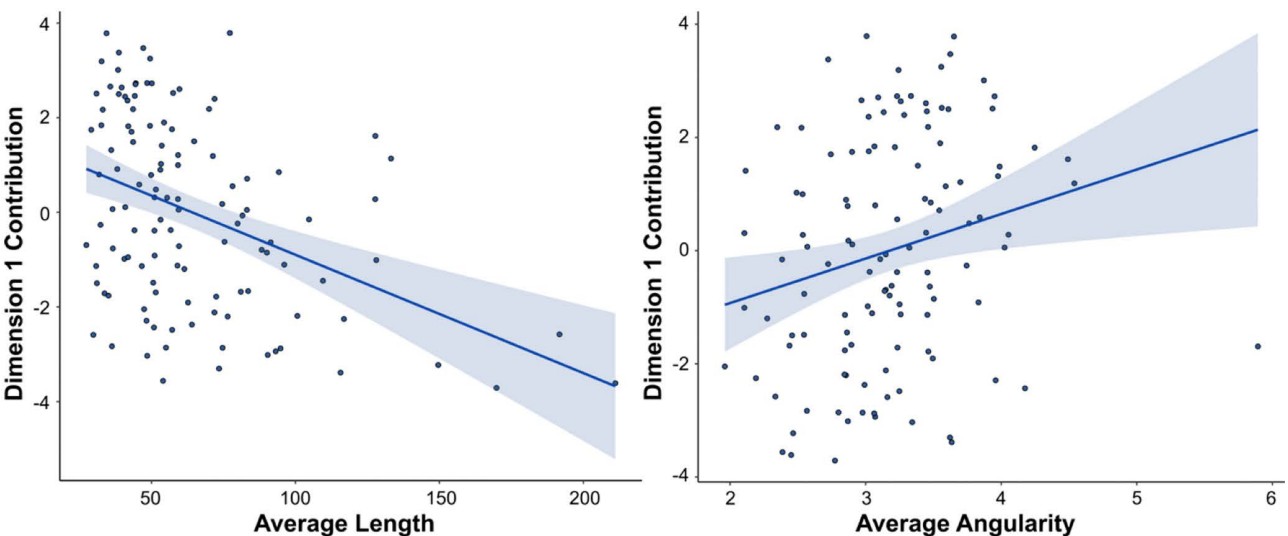

**Fig 10. Image feature regression plots for the two significant predictors of Dimension 1.** Length (left) is negatively related to this dimension while Angularity (right) is positively related.

bridge only. However, to verify this, we ran one final experiment with Database 2, i.e., 4 views of 50 bridges taken from different angles, for a total of 200 images (see Fig 11 for examples). All images were resized to 800 by 640 pixels. The final dataset containing the images, the four subjective ratings, and the annotated design features is provided in the BDA dataset.

## Participants

A total of 54 people participated in Experiment 3 (27 men, 20 women, 2 non-binary individuals; $M_{age}$ = 25 years, $SD_{age}$ = 4.33 years, age range = 18–35 years). Participants were recruited via Prolific and compensated 1.50 GBP for their participation. Recruitment and testing began on June 28[th], 2023 and ended on July 5[th], 2023. We screened for English-speaking Prolific users who were at least 18 years old, had normal or corrected-to-normal vision, and maintained

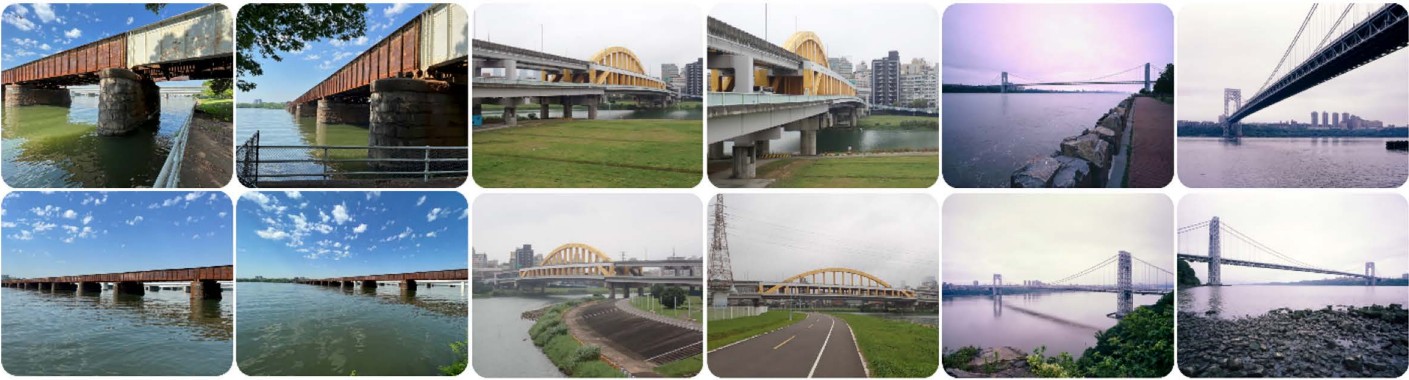

**Fig 11. Conceptual illustration of bridges' repetition.** *Each bridge is presented through both partial views and wide shots. The partial views highlight specific sections of the bridges, emphasizing key structural details, while the wide shots capture the bridges in their entirety, showing how they interact with their environmental surroundings.*

a minimum approval rate of 95%. To ensure adequate attention during the task, participants were required to press a specific key multiple times during the experiment. Those who failed these attention checks were excluded from the final sample. Additionally, participants who made the same response in more than 40 trials were excluded. To confirm that participants completed the experiment without rushing, we also excluded those whose completion time deviated by more than three standard deviations from the mean (approximately 30.2 minutes). Based on these exclusion criteria, 2 participants were excluded for failing the attention checks, 2 for exceeding the maximum repeated responses, and 1 for failing the completion time requirement.

## Procedure

Experiment 3 was programmed using jsPsych and was run on an individually hosted website. The image order was randomized to ensure participants would not view the same bridge sequentially. Under this design, each of the 200 images was rated once by each participant, resulting in 49 ratings per image.

Participants were presented with a digital consent form at the beginning of the experiment. Then, participants were asked to provide their age and gender in a demographics survey. We also collected participants' relevant background information similar to Experiment 1 and 2: (a) Do you have a background in architecture? (b) Do you have a formal education in architecture? (c) Do you have a background in engineering? (d) Do you have a formal education in engineering? Participants were given the option to choose between 0 = yes and 1 = no. Following the demographics survey, participants were shown with an instruction screen and were informed that they would be seeing bridge images and asked to evaluate the bridge design on various aspects. Participants would then proceed to the actual experiment by pressing the keyboard. Each trial started by displaying an image in the center of the screen on a light gray background. The 5-point Likert scale was presented beneath the image. We used the same scale to Experiment 1, where participants were asked to provide their response via mouse click on aesthetic liking, complexity, interest, and perceived safety (see above).

Participants were given a voluntary break every 200 trials. After the main experiment, participants were asked to indicate if they had encountered any technical difficulties, such as failed image loading. No technical difficulties were reported.

## Data analysis

We computed the average ratings of aesthetic liking, complexity, interest, and perceived safety based on participants' responses in the task. The ratings were firstly z-normalized for each participant and each task. Then, for each image, we averaged across all participants who viewed that image, resulting in one rating per image for each of the subjective measures. For each rating item, we excluded outliers by removing trials where the aggregated score was more than three standard deviations away from the mean. No outliers were removed based on this criterion.

To investigate whether participants were paying attention to the bridge design as opposed to other visible features in the images, we compared the variance of the ratings they gave for the same bridge with the variance of the ratings across bridges. If the within-bridge variance is smaller than the between-bridge variance, meaning that participants rated images of the same bridge more consistently than they rated different bridges, we would conclude that such bridge images have high consistency. We computed the within-bridge variance and across-bridge variance for all four rating items. Then, we conducted one-sample t-tests to verify if the within-bridge variance is smaller than the across-bridge variance.

## Results

Fig 12 shows the within-bridge variance and across-bridge variance for aesthetic, complexity, interest, and perceived safety ratings. The density plot displays the distribution of the bridges' within-bridge variance, and the dotted line is the across-bridge variance for each rating item. One sided t-test found that within-bridge variances are smaller than the across-bridge variance for all four rating items (aesthetic: $t(49) = -76.20$, $p < 0.001$; complexity: $t(49) = -96.55$, $p < 0.001$; interest: $t(49) = -33.52$, $p < 0.001$; perceived safety: $t(49) = -72.37$, $p < 0.001$). These results suggest that participants

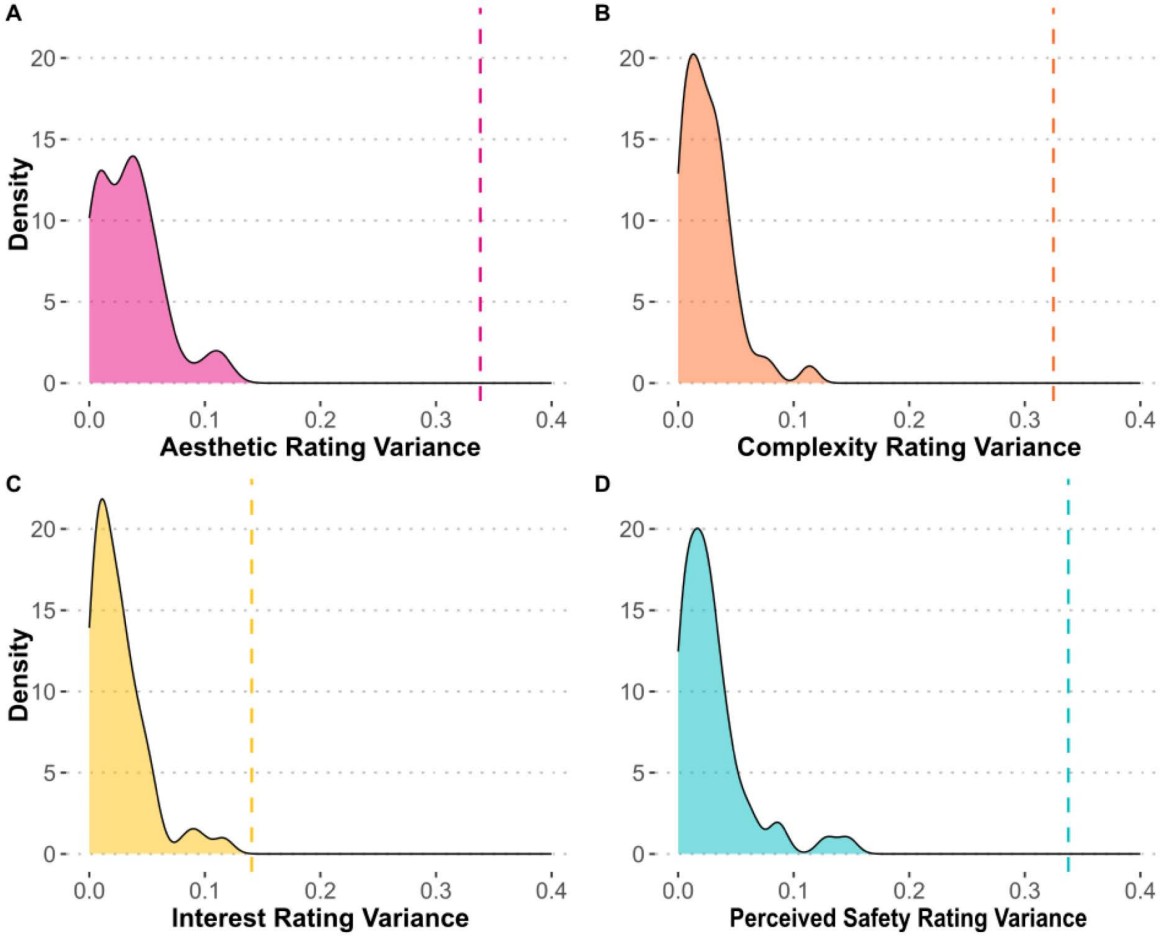

**Fig 12. Comparison of within-bridge and across-bridge rating variances across four rating scales. (A)** Comparison of within-bridge and across-bridge rating variances for aesthetic (pink). The density plot represents the within-bridge variance, while the dashed line indicates the across-bridge variance. **(B)** Comparison for complexity (orange). **(C)** Comparison for interest (yellow). **(D)** Comparison for perceived safety (blue). The visual comparison reveals that the within-bridge variances are consistently smaller than the across-bridge variance across all four rating scales.

provided consistent ratings for the same bridge and validate that the participants were attentively evaluating the bridge itself and not the background.

## Discussion

We explored the relationship between the aesthetic evaluations of bridges and their design features (i.e., visible features that determine function and economy). Using FAMD, we found two significant dimensions. The first dimension, which we named the *Aesthetic* dimension, related to the aesthetic pleasantness, interest, and complexity of a given bridge. These judgements depended largely on bridge type. Specifically, people found cable-stayed, suspension, and truss bridges more pleasant, interesting, and complex than slab, frame, and girder bridges.

The second dimension, which we named the *Perceived Safety* dimension, related to subjective judgements of safety, and were most dependent on the material and age of the bridges, with stone bridges from before the industrial revolution being rated as the safest. Except for one outlier, however, the perceived safety ratings across all bridges were relatively high, meaning that people generally believed the bridges to be safe. This is perhaps because they knew that the bridges

existed in the real world and therefore must be safe enough for public use. This is consistent with the exceptionally high degree of reliability generally provided by bridges both in the past and currently. While our study focused on subjective perceptions of safety, work in transportation research has linked features of the environment with measurable safety outcomes [34]. For example, roadway factors (e.g., speed limit), lighting (e.g., daylight), and structural elements (e.g., urban vs. rural) have been shown to influence crash risk and pedestrian injury. Future research could explore whether bridge features that are intuitively associated with safety are similarly linked to objective safety outcomes or incident rates. This connection between perceived and actual safety could inform both public communication and design practices for new infrastructure.

The pattern of results was apparent across two distinct participant samples (Experiments 1 and 2), demonstrating the robustness of our findings. Additionally, although we found that most of the bridges were not recognized as familiar by the majority of participants, we further removed any potential familiarity confound in the analysis of Experiment 2 to ensure that the aesthetic judgements relied solely on a bridge's visual appearance and not on any prior knowledge or emotional connection to the bridge.

Finally, we also wanted to ensure that the judgements were made only on the bridges themselves and not on the scenery or background present within the image. We did this primarily by instructing participants to focus only on the bridge when making their judgements. In a further image feature analysis, exploring how several visual features of either the bridge or the background related to the subjective ratings, we found that only the bridge features were predictive of the *Aesthetic* dimension, confirming that participants indeed followed instructions to focus only on the bridge within the image. As a final confirmation, we ran one more study using a new subset of images, comprising of bridges from several distinct viewpoints. This meant that a set of four images contained the same bridge to be rated, but included different foreground and background information depending on the viewpoint. Here we found that the consistency of ratings across images of the *same* bridge was much higher than across images of different bridges, suggesting that it was indeed the bridge that was being judged, and not other information about the scene.

Regarding the image feature analysis in particular, we found an intriguing result pertaining to angularity that runs counter to what is typically found in the literature on visual features and aesthetic pleasure. Angularity is typically negatively associated with liking or positive valence [18–20,35]. In fact, the "curvature effect" (i.e., the preference for smooth curves over sharp angles) is one of the most robust findings in empirical aesthetics [36]. Longer contours have also been associated with positive valence and safety [20] and positive emotions, such as joy and wonder [37]. In the current study, we found the opposite pattern, where high angularity and shorter contours were positively associated with aesthetic pleasure, interest, and complexity ratings. This certainly makes sense within the context of our image set, with cable-stayed and suspension bridges being considered the most pleasant, interesting, and complex types of bridges, and also being made up of more short and angular contours compared to slab bridges, due to their designs. However, within the broader field of empirical aesthetics, this atypical pattern of results may suggest that so called "structural art", which includes structures such as bridges, towers, and long-span roofs [6], may derive its aesthetic appeal from different factors compared to other forms of architecture (e.g., building façades, indoor rooms, etc.) or typical aesthetic objects (e.g., sculptures, paintings, home décor, etc.).

However, this interpretation should be made with care, as it is possible that the real-world size of a given bridge, which is determined primarily by its span length, is an important factor underlying these observations. Span length, which was not included as a design feature in this study, has a significant effect on the choice of bridge type in the design process. With very few exceptions, cable-stayed bridges and suspension bridges are used only for long-span bridges because they are generally not economical for short spans. When large bridges are captured in an image, the details of the design would necessarily shrink to fit within certain dimensions, and the graceful curves of the cables will be computed as short angular lines by computational analyses. Lay people, however, may be able to infer the true span and design of the bridge within the image from prior knowledge and experience. Large structures are known to inspire awe and wonder [38],

therefore, it is possible that the aesthetic responses were primarily due to a bridge's true size rather than the properties of computationally extracted visual features.

Additionally, all other factors being equal, as span length increases, so does the technical challenge faced by the engineers who design bridges and, consequently, so does the impetus for creative solutions to problems related to practical function and economy. Gauvreau [21] posited that the visual expression of creative ideas, even when these are conceived entirely to bring about improvements in practical function or economy, can evoke an aesthetic response, even among lay people who have no formal understanding of the underlying technical principles. So, it is likewise possible that high aesthetic ratings to cable-stayed and suspension bridges was in part due to their response to the visual expression of creative ideas embodied in these works. More work should be carried out to better understand how such factors may influence the aesthetics of bridges. For instance, follow-up studies to determine whether observers have the ability to infer span length from the 2D image, and whether that is related to their sense of awe, aesthetic pleasure, and perceived creativity of the design could distinguish between these possible explanations, or find that they are both valid to some extent.

Studying bridges in particular is important because bridges are extremely prevalent in our urban spaces that we interact with daily. Modern urban life poses many challenges to wellbeing [39,40]. The design of urban spaces has a strong effect on people's visual comfort and navigation abilities [41,42], stress [43], mood [44], and overall life satisfaction [45], suggesting that the aesthetics of urban infrastructure may be just as important as, and even intertwined with, its functionality. Our insights about which bridge types are preferred could be useful for informing design decisions in contexts where public engagement or visual impact are important considerations. For example, city planners might consider incorporating the preferred bridge types into highly visible or frequented areas within a city. Conversely, when visual impact is a secondary consideration, more economical or simpler bridge types may be more appropriate. Additionally, our findings on material and perceived safety may guide policy considerations around heritage preservation, since older stone bridges were perceived as safer and more appealing, potentially justifying their maintenance in urban planning contexts.

Continuing to explore how different aspects of urban design influence people's aesthetic experiences will be of utmost importance in society, as urbanization continues. To do so, collaborations between cognitive scientists who study the human experience, and the engineers and other design professionals who create our urban infrastructure, is imperative. For instance, some engineers' expert opinions about what makes a bridge aesthetically pleasing are not supported by our findings, since people tend to prefer complexity over simplicity or minimalism. However, relying too strongly on lay people's preferences may lead to unnecessary design additions just for the sake of appearance [46] which could have disastrous consequences (e.g., Miami pedestrian bridge collapse) [47]. A true collaborative effort between the empirical aesthetics and engineering fields is encouraged. It has the potential to bring about a positive transformation of urban design and planning policy by improving the balance between the aesthetic preferences of design professionals and the public.

Our study was conducted on groups of non-expert participants, thus we can only draw conclusions on the relationship between aesthetics and bridge design for the average person. Future work could explore how this relationship might change depending on expertise by testing different expert groups, such as engineers, architects, and other relevant design professionals. Art knowledge and expertise has been shown to influence aesthetic judgements [48,49], so it is possible that the correlations between aesthetic judgements and design features may differ from the current findings. It is an open question, for instance, to what extent experts' aesthetic evaluations are related to purely visual features vs. design choices such as non-functional ornamentation (i.e., aesthetic premium). Additionally, while our image set includes bridges from around the world, and our participants reside in countries around the world as well (see S1 Text), we lack sufficient data for comparisons of aesthetic evaluations across cultures. Cultural context has been shown to shape aesthetic responses to artworks [50] and everyday objects [51], thus follow-up studies should explore whether individuals from different cultural regions evaluate bridge aesthetics differently. Understanding how cultural context interacts with visual and structural design features will be crucial for improving the generalizability and cross-cultural applicability of our findings.

We provide our image set and data as a resource for both the empirical aesthetics and bridge engineering communities, in hopes of fostering further collaborations and inspiring future research on the aesthetics of bridges and other urban infrastructure. Although we believe our dataset in its current form will be useful, it is only a start. One limitation of the current dataset is that the categorical variables are unbalanced. That is, we do not have an equal number of images representing each bridge type or other design features, nor is it possible to have each combination of design features represented. For instance, arch bridges, by definition, do not have a constant depth, and bridges built before the industrial revolution were not made of steel. This is, in part, why we used FAMD to analyze our data, to uncover the relationships that exist between the categorical variables. Still, unbalanced variables impose a stronger influence on FAMD, leading to a potentially inflated impact of factors such as visible material or apparent age. Thus, we interpret our findings, especially of the *Perceived Safety* dimension, with caution. This limitation highlights that much work can and should be done to expand the dataset in order to further explore unique and novel questions. For example, given that the 5 dimensions found in the FAMD account for approximately 50% of explained variance, there are certainly other factors influencing bridge aesthetics that we did not analyze.

Overall, our work provides the first empirical study of the aesthetics of bridge design, offering a foundational understanding of how specific design features that determine function and economy relate to the visual appeal of bridges. Bridges are more than just functional infrastructure; they can become symbols that reflect cultural values and enhance the beauty of our built environment. To enable further exploration of these themes, we offer a freely available data set to be used and expanded upon by both the empirical aesthetics and engineering communities. As we have demonstrated here, collaborations between distinct fields enable us to address novel questions that resonate not only with the collaborating disciplines, but also with broader audiences beyond these scientific communities. Understanding the factors that make bridges, and other works of public infrastructure, aesthetically pleasing will inform future urban design initiatives that have the potential to improve the wellbeing of society.

## Supporting information

**S1 Fig. Scree Plot for Experiment 1.**
(PDF)

**S2 Fig. Scree Plot for Experiment 2.**
(PDF)

**S3 Fig. Correlation Matrix of Aesthetic Ratings for Experiment 1.**
(PDF)

**S4 Fig. Correlation Matrix of Aesthetic Ratings of Experiment 2.**
(PDF)

**S5 Fig. Familiarity Distribution of Bridges in Experiment 1 and 2.**
(PDF)

**S1 Table. Contribution of Variables to Dimension 1 and 2 of Experiment 1.**
(PDF)

**S2 Table. Contribution of Variables to Dimension 1 and 2 of Experiment 2.**
(PDF)

**S1 Text. Participant Demograghics, Professional Background, and Country of Residence.**
(PDF)

## Author contributions

**Conceptualization:** Mei Yang, Paul Gauvreau, Dirk B. Walther.

**Data curation:** Mei Yang.

**Formal analysis:** Mei Yang, Claudia Damiano.

**Funding acquisition:** Dirk B. Walther.

**Resources:** Paul Gauvreau.

**Supervision:** Claudia Damiano, Paul Gauvreau, Dirk B. Walther.

**Visualization:** Mei Yang, Claudia Damiano.

**Writing – original draft:** Mei Yang, Claudia Damiano, Paul Gauvreau.

**Writing – review & editing:** Mei Yang, Claudia Damiano, Paul Gauvreau, Dirk B. Walther.

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
