## [Decision Letter · Decision Letter 0]

25 Sep 2025

PONE-D-25-31773Empirical aesthetics of bridgesPLOS ONE

Dear Dr. Damiano,

Thank you for submitting your manuscript to PLOS ONE. After careful consideration, we feel that it has merit but does not fully meet PLOS ONE’s publication criteria as it currently stands. Therefore, we invite you to submit a revised version of the manuscript that addresses the points raised during the review process.

**Most of the reviewers find the manuscript sound and of interest, with suggestions for minor revisions to improve clarity and completeness. I therefore recommend minor revisions as detailed in the reviewers’ comments.**

We look forward to receiving your revised manuscript.

Kind regards,

Matteo Bodini, Ph.D.

Academic Editor

PLOS ONE

**Journal Requirements:**

1. When submitting your revision, we need you to address these additional requirements. Please ensure that your manuscript meets PLOS ONE's style requirements, including those for file naming. The PLOS ONE style templates can be found at https://journals.plos.org/plosone/s/file?id=wjVg/PLOSOne_formatting_sample_main_body.pdf and https://journals.plos.org/plosone/s/file?id=ba62/PLOSOne_formatting_sample_title_authors_affiliations.pdf 2. Please note that PLOS ONE has specific guidelines on code sharing for submissions in which author-generated code underpins the findings in the manuscript. In these cases, we expect all author-generated code to be made available without restrictions upon publication of the work. Please review our guidelines at https://journals.plos.org/plosone/s/materials-and-software-sharing#loc-sharing-code and ensure that your code is shared in a way that follows best practice and facilitates reproducibility and reuse. 3. Thank you for stating in your Funding Statement: This work was supported by a Discovery Grant (RGPIN-2020-04097) from the Natural Sciences and Engineering Research Council of Canada (https://www.nserc-crsng.gc.ca/) and an Insight Grant (435-2023-0015) from the Social Sciences and Humanities Research Council (https://sshrc-crsh.canada.ca/) awarded to DBW. The funders did not play a role in the study design, data collection and analysis, decision to publish, or preparation of the manuscript.  Please provide an amended statement that declares *all* the funding or sources of support (whether external or internal to your organization) received during this study, as detailed online in our guide for authors at http://journals.plos.org/plosone/s/submit-now.  Please also include the statement “There was no additional external funding received for this study.” in your updated Funding Statement. Please include your amended Funding Statement within your cover letter. We will change the online submission form on your behalf. 4. We note that Figures 1, 2, 3 and 11 in your submission contain copyrighted images. All PLOS content is published under the Creative Commons Attribution License (CC BY 4.0), which means that the manuscript, images, and Supporting Information files will be freely available online, and any third party is permitted to access, download, copy, distribute, and use these materials in any way, even commercially, with proper attribution. For more information, see our copyright guidelines: http://journals.plos.org/plosone/s/licenses-and-copyright. We require you to either present written permission from the copyright holder to publish these figures specifically under the CC BY 4.0 license, or remove the figures from your submission: a. You may seek permission from the original copyright holder of Figures 1, 2, 3 and 11 to publish the content specifically under the CC BY 4.0 license.  We recommend that you contact the original copyright holder with the Content Permission Form (http://journals.plos.org/plosone/s/file?id=7c09/content-permission-form.pdf) and the following text:“I request permission for the open-access journal PLOS ONE to publish XXX under the Creative Commons Attribution License (CCAL) CC BY 4.0 (http://creativecommons.org/licenses/by/4.0/). Please be aware that this license allows unrestricted use and distribution, even commercially, by third parties. Please reply and provide explicit written permission to publish XXX under a CC BY license and complete the attached form.” Please upload the completed Content Permission Form or other proof of granted permissions as an "Other" file with your submission.  In the figure caption of the copyrighted figure, please include the following text: “Reprinted from [ref] under a CC BY license, with permission from [name of publisher], original copyright [original copyright year].” b. If you are unable to obtain permission from the original copyright holder to publish these figures under the CC BY 4.0 license or if the copyright holder’s requirements are incompatible with the CC BY 4.0 license, please either i) remove the figure or ii) supply a replacement figure that complies with the CC BY 4.0 license. Please check copyright information on all replacement figures and update the figure caption with source information. If applicable, please specify in the figure caption text when a figure is similar but not identical to the original image and is therefore for illustrative purposes only. 5. Please include captions for your Supporting Information files at the end of your manuscript, and update any in-text citations to match accordingly. Please see our Supporting Information guidelines for more information: http://journals.plos.org/plosone/s/supporting-information. 6. If the reviewer comments include a recommendation to cite specific previously published works, please review and evaluate these publications to determine whether they are relevant and should be cited. There is no requirement to cite these works unless the editor has indicated otherwise. 

Reviewers' comments:

Reviewer's Responses to Questions

**Comments to the Author**

1. Is the manuscript technically sound, and do the data support the conclusions?

Reviewer #1: Yes

Reviewer #2: Yes

Reviewer #3: Yes

2. Has the statistical analysis been performed appropriately and rigorously?

Reviewer #1: Yes

Reviewer #2: No

Reviewer #3: Yes

3. Have the authors made all data underlying the findings in their manuscript fully available?

Reviewer #1: Yes

Reviewer #2: No

Reviewer #3: Yes

4. Is the manuscript presented in an intelligible fashion and written in standard English?

Reviewer #1: Yes

Reviewer #2: No

Reviewer #3: Yes

5. Review Comments to the Author

**Reviewer #1: ** Summary

This manuscript presents a comprehensive, well-designed, and highly original empirical investigation of the aesthetic perception of bridges. Drawing from a dataset of 318 real-world bridges, rated by over 250 participants across multiple dimensions (aesthetic liking, complexity, interest, perceived safety), the authors identify two latent perceptual dimensions—Aesthetic and Perceived Safety—and examine how they relate to both structural design features and mid-level visual properties. The study integrates civil engineering expertise with cognitive science methods and contributes a publicly available dataset for future research.

The topic is timely and underexplored, and the study is methodologically sound and theoretically significant. The writing is clear, the analyses are appropriate and replicable, and the discussion is thoughtful and well contextualized.

Major Comments

• The manuscript explains that design features were annotated by experts, but it is not clear whether one or multiple annotators performed the coding. If multiple coders were involved, inter-rater reliability (e.g., Cohen’s κ) should be reported. If a single expert conducted the annotations, this should be stated explicitly, along with a brief justification (e.g., relevant experience or qualifications).

• While the authors rightly emphasize that their study captures laypeople’s perspectives, it would be helpful to acknowledge that aesthetic preferences may vary across cultures. The discussion could be strengthened by briefly addressing the potential role of cultural context—e.g., whether individuals from Asia, Europe, or Africa might evaluate bridge aesthetics differently—and what that might imply for the generalisability of the findings.

• The finding that angularity positively predicts aesthetic responses contradicts a large body of literature in empirical aesthetics. While the authors offer plausible and insightful explanations (e.g., domain-specific aesthetics of structural engineering, inferred size and complexity), these remain speculative. The discussion could be enhanced by suggesting concrete follow-up experiments to disentangle these interpretations.

Minor Comments:

• Missing p-value (line 518): In reporting the results of Experiment 3, the p-value for interest is missing: “interest: t(49) = -33.52;”

Please add the corresponding p-value for consistency (presumably p < 0.001).

• Typographical error (line 618): The word “For” is incorrectly capitalised in the middle of the sentence

• While the diversity of bridges is mentioned, a brief breakdown of bridge distribution by type, material, age, aesthetic premium and region (if available) would help readers assess the representativeness of the dataset.

Conclusion

This is an important and methodologically rigorous contribution to the growing field of empirical aesthetics. It advances our understanding of how large-scale engineered structures are perceived, opens new directions for interdisciplinary research, and provides a high-quality dataset for future studies. With minor revisions as suggested above, I strongly recommend this manuscript for publication.

**Reviewer #2: ** REJECTED. This is trivial disorganized and immature work both technically/analytically and structurally. Research work without ‘conclusion’???

The lack of innovation of this work in the science border is concrete!! The main novelty?? characterized contribution??? It simply can be seen from superficial review and all outdated references, with at least 3 years gap!!!!

There are many untreated technical concerns: The framework is a simple straight image analysis without any clear modeling procedure/proper justifications/lacking substantive enhancements to bolster the theoretical underpinnings and methodological robustness of the study/feature extraction strategy, noise removal/deep discussion about the claimed ‘dimension’, ‘safety’…

The prerequisite of this draft is the use of a unified pixel size and therefore, any modification to input data such as changes in resolution (pixels, grid structure, …) compromises data integrity. Hence, the capability and stability approval of the given model is not certified.

Incohesive English/inconsistent with the third passive/frequent flaws…

Unconvincing and documented Discussion/technical limitations/solid comparison with other scholars’ approaches/constraints in getting features/…

If and only if we consider Fig 10, it is simple uncertainty quantification where in comparison with advanced approaches like https://link.springer.com/article/10.1007/s00366-023-01852-5... It is uncompetitive

Ill-formatted references in terms of required identifiers

**Reviewer #3: ** Clarity of Methods: The paper is well structured and clearly written. One area that could use a bit more detail is the description of the Factorial Analysis of Mixed Data (FAMD). A brief explanation of why FAMD was chosen over other dimensionality reduction methods would help readers less familiar with the technique.

Participant Characteristics: The study includes ratings from 254 participants, which is a strong sample size. It would strengthen the paper to provide a more detailed demographic breakdown (e.g., cultural or professional background), since perceptions of aesthetics and safety can vary across populations.

Practical Implications: The findings on bridge type and material are compelling. Expanding the discussion slightly on how these insights might inform future design practices or policy guidelines could enhance the paper’s practical relevance.

Perceived vs. Actual Safety : Since perceived safety emerged as a key dimension, it may be helpful to briefly connect this with research on actual safety outcomes. For example, 10.1080/23249935.2025.2516817. While focused on road safety, it shows how infrastructure features translate into measurable safety risks. Referencing such work could broaden the discussion by linking perceptions with outcome-based evidence.

6. PLOS authors have the option to publish the peer review history of their article (what does this mean? ). If published, this will include your full peer review and any attached files.

**Do you want your identity to be public for this peer review?** For information about this choice, including consent withdrawal, please see our Privacy Policy .

Reviewer #1: **Yes: ** Ivan Stojilovic

Reviewer #2: No

Reviewer #3: No

---

## [Author Response · Author response to Decision Letter 1]

12 Nov 2025

See Response to Reviewers document.

---

## [Decision Letter · Decision Letter 1]

25 Nov 2025

Empirical aesthetics of bridges

PONE-D-25-31773R1

Dear Dr. Damiano,

We’re pleased to inform you that your manuscript has been judged scientifically suitable for publication and will be formally accepted for publication once it meets all outstanding technical requirements.

Kind regards,

Matteo Bodini, Ph.D.

Academic Editor

PLOS ONE

Additional Editor Comments (optional):

Reviewers' comments:

Reviewer's Responses to Questions

**Comments to the Author**

1. If the authors have adequately addressed your comments raised in a previous round of review and you feel that this manuscript is now acceptable for publication, you may indicate that here to bypass the “Comments to the Author” section, enter your conflict of interest statement in the “Confidential to Editor” section, and submit your "Accept" recommendation.

Reviewer #1: All comments have been addressed

Reviewer #3: All comments have been addressed

2. Is the manuscript technically sound, and do the data support the conclusions?

Reviewer #1: Yes

Reviewer #3: Yes

3. Has the statistical analysis been performed appropriately and rigorously?

Reviewer #1: Yes

Reviewer #3: Yes

4. Have the authors made all data underlying the findings in their manuscript fully available?

Reviewer #1: Yes

Reviewer #3: (No Response)

5. Is the manuscript presented in an intelligible fashion and written in standard English?

Reviewer #1: Yes

Reviewer #3: (No Response)

6. Review Comments to the Author

Reviewer #1: I appreciate the thorough revisions made by the authors in response to my comments. All the concerns I raised in my initial review—including clarity in terminology, cautious interpretation of results, improvements to figure legends and table formatting, and clearer explanations of the modeling assumptions—have been adequately addressed. The manuscript is now more balanced, transparent, and accessible to a broader audience.

I find the revised version acceptable for publication and have no further comments.

Reviewer #3: My comments on the previous version of the manuscript has been addressed. Goodluck with your future research

7. PLOS authors have the option to publish the peer review history of their article (what does this mean? ). If published, this will include your full peer review and any attached files.

**Do you want your identity to be public for this peer review?** For information about this choice, including consent withdrawal, please see our Privacy Policy .

Reviewer #1: **Yes: ** Ivan Stojilović

Reviewer #3: No

---

## [Editor Report · Acceptance letter]

PONE-D-25-31773R1

PLOS One

Dear Dr. Damiano,

I'm pleased to inform you that your manuscript has been deemed suitable for publication in PLOS One. Congratulations! Your manuscript is now being handed over to our production team.

Kind regards,

on behalf of

Dr. Matteo Bodini

Academic Editor

PLOS One